# The Serine/Threonine Kinase AP2-Associated Kinase 1 Plays an Important Role in Rabies Virus Entry

**DOI:** 10.3390/v12010045

**Published:** 2019-12-30

**Authors:** Chong Wang, Jinliang Wang, Lei Shuai, Xiao Ma, Hailin Zhang, Renqiang Liu, Weiye Chen, Xijun Wang, Jinying Ge, Zhiyuan Wen, Zhigao Bu

**Affiliations:** 1State Key Laboratory of Veterinary Biotechnology, Harbin Veterinary Research Institute, Chinese Academy of Agricultural Sciences, Harbin 150069, China; wangchong@caas.cn (C.W.); wangjinliang@caas.cn (J.W.); shuailei@caas.cn (L.S.); maxiao0605@126.com (X.M.); zhanghl0523@163.com (H.Z.); liurenqiang@caas.cn (R.L.); chenweiye@caas.cn (W.C.); wangxijun@caas.cn (X.W.); gejinying@caas.cn (J.G.); 2Jiangsu Co-innovation Center for Prevention and Control of Important Animal Infectious Diseases and Zoonoses, Yangzhou University, Yangzhou 225009, China

**Keywords:** rabies, AAK1, AP2M1, sunitinib

## Abstract

Rabies virus (RABV) invades the central nervous system and nearly always causes fatal disease in humans. RABV enters cells via clathrin-mediated endocytosis upon receptor binding. The detailed mechanism of this process and how it is regulated are not fully understood. Here, we carried out a high-through-put RNAi analysis and identified AP2-associated kinase 1 (AAK1), a serine/threonine kinase, as an important cellular component in regulating the entry of RABV. AAK1 knock-down greatly inhibits RABV infection of cells, and AAK1-induced phosphorylation of threonine 156 of the μ subunit of adaptor protein 2 (AP2M1) is found to be required for RABV entry. Inhibition of AAK1 kinase activity by sunitinib blocked AP2M1 phosphorylation, significantly inhibiting RABV infection and preventing RABV from entering early endosomes. In vivo studies revealed that sunitinib prolongs the survival of mice challenged with RABV street virus. Our findings indicate that AAK1 is a potential drug target for postexposure prophylaxis against rabies.

## 1. Introduction

Rabies is a fatal zoonotic disease that leads to nearly 60,000 deaths annually worldwide [1]. It is caused by the rabies virus (RABV), which is highly neurotropic. Human infections are usually initiated by bites from RABV-infected animals. RABV has a very broad host spectrum, and almost all warm-blooded animals can be infected. Once symptoms appear, no treatment has been proven to prevent death, and the mortality rate is almost 100% [2]. The mystery of RABV neuropathogenesis is still being unraveled despite many years of research, and this lack of information has hindered the development of an effective therapy for rabies.

RABV belongs to the genus *Lyssavirus* of the family Rhabdoviridae. The single-strand, negative-sense genome of the bullet-shaped virus encodes five genes: nucleoprotein (N), phosphoprotein (P), matrix protein (M), glycoprotein (G), and large polymerase protein (L). G exists as a trimer that embeds in the viral envelope, functioning in receptor binding and cell membrane penetration via low pH-induced membrane fusion. The G protein also determines the neurotropism of RABV. RABV uses multiple receptors to invade cells [3,4]. Upon receptor binding, RABV enters the cell through clathrin-mediated endocytosis (CME) [5]. Subsequently, the RABV-containing endosomes are transported through the cellular endosomal compartment to the late endosomes, where RABV enters the cytosol [6,7]. CME is a fundamental process that is commonly used by cells to engulf membrane-associated cargo proteins. Clathrin adaptors, which function as scaffolds between the clathrin lattice and the cargo protein, play a pivotal role in CME. There are many proteins reported as clathrin adaptors involved in clathrin-mediated endocytosis from the plasma membrane, including amphiphysins 1 and 2, AP2 complex, ARH protein, β-arrestin, Dab2, and epsin 1 [8]. Additional clathrin adaptors, such as the AP1 complex and GGA proteins, are involved in trafficking between the trans-Golgi network (TGN) and the endosomes.

There are four APs in mammalian cells—AP1, AP2, AP3, and AP4—that are heterotetramers of around 300 kDa. Each carries out similar functions in binding to the cargo motif, membrane components (mainly phosphoinositide), clathrin, and other accessory proteins involved in clathrin-coated vesicle (CCV) formation, and each is subjected to regulation by phosphorylation [8]. AP2 comprises four subunits: α (AP2A1), β (AP2B1), μ (AP2M1), and σ (AP2S1). AP2 recognizes and binds to the cytoplasmic sorting motif of the transmembrane cargo protein. At the same time, AP2 also binds to clathrin to form clathrin-coated pits (CCPs). After maturation, the cargo-containing CCPs are pinched off by dynamin from the plasma membrane and transported through cellular endosomal organelles [9,10].

In this study, we carried out a high-through-put (HPT) RNAi assay in attempt to identify cellular factors that affects RABV entry into the cells. Through these analyses, we found that AP2-associated kinase 1 (AAK1) plays a critical role in regulating the clathrin-mediated endocytosis of RABV. Knock-down of AAK1 significantly decreased the infection of cells by RABV. Further analysis revealed that the phosphorylation of AP2M1 by AAK1 is critical for RABV entry. Moreover, the sunitinib, an FDA-approved receptor tyrosine kinase (RTK) inhibitor that effectively inhibits the AAK1 kinase, significantly suppressed RABV infection of cells in vitro, and also prolonged the survival of mice challenged with RABV street virus in vivo. Our results demonstrate that AAK1 is a potential drug target for RABV, and that sunitinib could be a useful drug for RABV postexposure prophylaxis.

## 2. Materials and Methods

### 2.1. Ethics Statement

The animal experiments in this study were approved by the Committee on the Ethics of Animal Experiments of the Harbin Veterinary Research Institute of the Chinese Academy of Agricultural Sciences. Mice were housed and handled in accordance with the Guide for the Care and Use of Laboratory Animals of the Ministry of Science and Technology of China. The RABV challenge study was conducted in the biosafety level 3 (BSL-3) facilities in the Harbin Veterinary Research Institute of the Chinese Academy of Agricultural Sciences (CAAS) (approval no. IACUC-2017-080).

### 2.2. Cells and Virus

HEK-293 (ATCC CRL-1573) cells were maintained in DMEM supplemented with 10% FCS, L-glutamine, and penicillin-streptomycin. Human neuroblastoma cells SK-N-SH (SK cells) (ATCC HTB-11) were maintained in MEM/EBSS supplemented with 10% FCS, L-glutamine, and 1% penicillin-streptomycin. BSR-T7/5 cells were maintained in DMEM supplemented with 5% FCS, L-glutamine, and 1% penicillin-streptomycin. Mouse primary neuron (mPN) cells were generated from newborn mice. The mouse brain cortex was dissected and cut into small pieces, dissociated by trypsinization for 7 min at 37 °C, after which 100 µg of DNase was added to the cells for another 1 min. After centrifugation at 1000 rpm for 5 min, the cell pellet was dispersed and filtered through a cell strainer (BD Falcon, CA, USA). Further, mPN cells were cultured in Neurobasal media (ThermoFisher, Waltham, MA) supplemented with B27 (ThermoFisher, Waltham, MA), 2 mM glutamine, 25 mM HEPES, and 25 µg/mL ß-D-arabinofuranoside.

The RABV ERA strain was propagated, titrated, and maintained in our laboratory [11]. ERA-mCherry is a recombinant ERA that has the *mCherry* gene inserted between the ERA *M* and *G* genes as an additional transcription unit. The ERA-N-mCherry, a recombinant ERA with a gene inserted between the ERA *M* and *G* genes as an additional transcription unit that encodes for an ERA *N*-mCherry fusion protein, was generated as previously described [12]. The rabies street virus GX09 was passaged in the mouse brain and titrated to determine the intramuscular 50% lethal dose (LD_50_) in mice before the challenge study.

### 2.3. RNA Interference (RNAi)

Small interfering RNAs (siRNAs) were transfected into cells using RNAiMAX reagent (ThermoFisher, Waltham, MA, USA) at 48 h prior to rabies virus infection. Ambion Silencer Selected siRNA (ThermoFisher, Waltham, MA, USA) was used. Two AAK1 siRNA sequences were used: AAK1-1, CGUGAGUAGCGGUGAUGUAtt; AAK1-2 siRNA, GACAAGCAAUGGGAUGAAAtt; one AP2M1 siRNA sequence, AGUUUGAGCUUAUGAGGUAtt; and one RABV L siRNA sequence, GGAAUGCACUUUCGAUAUAtt; one GAK siRNA sequence, CGAGGAAUACAACACCAAUtt was used. A non-targeting (NT) siRNA (Ambion catalog no. 4390843) was used as a negative control. A total of 5 × 10^3^ HEK293 or SK cells seeded in 96-well plates per well were transfected in triplicates with AAK1, AP2M1 or GAK siRNA. After incubation for 48 h, siRNA knock-down cells were infected with ERA-mCherry, at an MOI of 0.1. At 48-h post-infection, cells were fixed with 4% paraformaldehyde and stained with Hoechst 33342 (Invitrogen, OR, USA) for 1 h. Cells were imaged and analyzed by using the PerkinElmer Operetta high-content system (PerkinElmer, Waltham, MA, USA) and Columbus software (PerkinElmer, Waltham, MA, USA). Fifty-two fields per well were imaged at 20× magnification. The images fluorescent cells and cell nuclei were automatically identified and quantified by software. The relative infection ratio was determined according to the number of infected cells versus the total number of cells per well. The assay was independently repeated three times.

### 2.4. qRT-PCR

Quantitative real-time (qRT) PCR was performed using Applied Biosystems SYBR Green Mix (ThermoFisher, Waltham, MA, USA). GAPDH was used as an internal control. Relative quantification was performed by the cycle threshold (ΔΔCT) method. The sequences for the AAK1-specific primers are forward primer 5′-CCACAAACTGAGGGAGTCAATGC-3′ and reverse primer 5′-ATGTCTGCCTTCGTAGTGATGATTTT-3′. The sequences for the RABV N-specific primers are forward primer 5′-ACTAGGCTTGAGTGGGAAATC-3′ and reverse primer 5′-GGAGCACATGCAGCAATAAC-3′.

### 2.5. Western Blotting

Cells were lysed with PBS containing 1% (v/v) NP-40, and subjected to Western blot analyses. The primary antibody for AAK1 was rabbit anti-AAK1 (ab134971, Abcam, Cambridge, UK) and that for phosphorylated AP2M1 was rabbit anti-phospho-AP2M1 antibody (ab109397, Abcam, Cambridge, UK) in the presence of 100 nM Calyculin A (Cell Signaling Technology, MA, USA). The primary antibody for GAK was rabbit anti-GAK (sc-137053, Santa Cruz Biotechnology, CA, USA). Horseradish peroxidase (HRP)-conjugated anti-rabbit IgG (A00098, GenScript, Nanjing, China) was used as the secondary antibody. Na+/K+ ATPase (ab76020, Abcam, Cambridge, UK) was used as a loading control. Band intensity was quantified by using ImageJ software.

### 2.6. Cell Viability Assay

Cell viability was determined by using a CellTiter-Glo kit (Promega, Madison, WI, USA). Briefly, HEK293, SK, or mPN cells were seeded onto 96-well plates with opaque walls. At 48 h after drug treatment or siRNAs transfection, cells were lysed for 1 h at room temperature in 100 μL of CellTiter-Glo reagent (Promega, Madison, WI, USA). Luminescence was measured with a GloMax 96 Microplate Luminometer (Promega, Madison, WI, USA).

### 2.7. Virus Titration

Twenty-four-well plates were seeded with BSR-T7/5 cells and incubated as described above. Then, the cells were infected with 10-fold serial dilutions of the virus. After incubation for 48 h, the number of focus-forming units (FFU) was counted in 24-well plates, and the titer was expressed as the reciprocal of the highest dilution titer.

### 2.8. Cell plasma Membrane Protein Isolation

Cell plasma membrane protein was isolated from 2.5 × 10^6^ cells by using the Minute plasma membrane protein isolation and cell fractionation kit (SM-005, Invent Biotechnologies, Plymouth, MN, USA) and following the manufacturer’s instructions. The pellet containing plasma membrane proteins was collected to analyze for AP2M1 (TA503028, Origene, Rockville, USA) and phosphorylated AP2M1 expression by Western blotting. Na+/K+ ATPase was used as the plasma membrane loading control.

### 2.9. Pharmacological Inhibition

HEK293, SK, or mPN cells were seeded onto 96-well plates. Cells were treated with various concentrations of the sunitinib (Sigma-Aldrich, MO, USA) or erlotinib (LC Laboratories, USA) or with dimethyl sulfoxide (DMSO) for 1 h at 37 °C before infection with ERA-mCherry at an MOI of 0.01. At 48 h post-infection, a high-content quantitative image-based analysis was used to measure the relative infection ratio. 

### 2.10. Multiplex Immunofluorescence

HEK-293 cells were treated with 2 μM sunitinib or DMSO for 1 h at 37 °C. Then, the cells were infected with ERA-N-mCherry (MOI = 50) at 4 °C for 1 h to allow binding. The cells were then immediately shifted to 37 °C for 10 min to allow internalization. The cells were then washed and fixed with 4% paraformaldehyde. After permeabilization (0.25% Triton X-100 in PBS for 15 min) and blocking (ZLI-9056, Zsbio, Beijing, China), the samples were incubated with the primary antibodies followed by the HRP-conjugated secondary antibodies. Tyramide Signal Amplification (TSA) reagent diluted to 1% in reaction buffer (NEL811001KT, PerkinElmer, Waltham, MA, USA) was added and incubated for at most 1 min until the best signal intensity and signal-to-noise ratio was achieved. Antibodies were then removed by incubation with stripping buffer at 37 °C for 30 min. Other antibodies against specific antigens were serially detected using spectrally different TSA reagents following the above method. The primary antibodies used in this study were: mouse anti-AP2M1 antibody, rabbit anti-Rab5 antibody (3547, Cell Signaling Technology, MA, USA), and rabbit anti-mCherry antibody (ab183628, Abcam, Cambridge, UK). HRP-conjugated anti-mouse IgG (A00160, GenScript, Nanjing, China) and HRP-conjugated anti-rabbit IgG (A00098, GenScript, Nanjing, China) were used as the secondary antibodies. Images were acquired using a Zeiss LSM880 laser-scanning confocal microscope (Carl Zeiss AG, Jena, Germany) with Airyscan (Plan-Apochromat, objective 63×, Numerical Aperture 1.4). Data were processed using Bitplane Imaris software (Bitplane AG, Zurich, Switzerland).

### 2.11. Mice Challenge Study

Groups of 10 five-week-old female C57BL/6 (B6) mice were purchased from Vital River Laboratories (Vital River Laboratories, Beijing, China). Group I was intramuscularly inoculated with 10 MLD_50_ of RABV street virus GX09. Group II was intramuscularly administered a daily total dose of 0.015 mg of sunitinib in 0.1 mL of PBS per mouse for five days. Group III was intramuscularly inoculated with 10 MLD_50_ of GX09 and then intramuscularly administered 0.015 mg of sunitinib per mouse 7 h after GX09 challenge; mice were given the same doses of sunitinib via the same route for the following 4 days. Mice were observed for 21 days for any signs of sickness or death. Survival rates were generated using GraphPad Prism software. Statistical significance was analyzed using the software built-in log-rank (Mantel-Cox) test.

## 3. Results

### 3.1. AAK1 Facilitates RABV Infection

A human siRNA library comprising 64,755 siRNAs and targeting 21,585 mRNAs in the human genome was used to identify potential host factors affecting RABV infection in the human embryonic kidney cell line HEK293. We found that downregulation of AAK1 mRNA significantly inhibited RABV infection. To confirm that AAK1 is necessary for RABV infection, we knocked-down AAK1 expression by transfecting HEK293 cells with two independent AAK1-targeting siRNAs, AAK1-1 and AAK1-2. A reduction of AAK1 protein expression at 48 h post-transfection was achieved, as shown by Western blotting (Figure 1A). Additionally, quantitative RT-PCR demonstrated that AAK1 mRNA expression was reduced by 84–85% in AAK1 siRNA-transfected HEK293 cells (Figure 1B). A growth kinetic assay with recombinant RABV ERA-mCherry (*mCherry* gene inserted between ERA *M* and *G* gene as an additional transcription unit) showed that the AAK1 knock-down inhibited RABV infection in HEK293 cells post-transfection (Figure 1C). We also confirmed that AAK1 siRNA knock-down inhibited the infection and spread of ERA-mCherry in HEK293 cells (Figure 1D,E) and in the human neuroblastoma cell line SK-N-SH (SK) by 42%–49% and 27%–31%, respectively (Figure 1F,G). The viability of HEK293 and SK cells was unaffected by AAK1 siRNA silencing when compared to the non-targeting siRNA (Figure 1E,G). Interestingly, the overexpression of AAK1 also decreased the ERA-mCherry growth titer (Figure 1H). These results indicate that AAK1 is important for RABV infection in vitro.

### 3.2. AAK1 Kinase Activity Facilitates RABV Infection

AAK1 is a member of the Prk/Ark family of serine/threonine kinases [13]. Sunitinib is an FDA-approved anti-cancer drug [14] that inhibits AAK1 kinase activity by binding to AAK1 with high affinity [15]. To determine whether AAK1 kinase activity plays a role in RABV infection, HEK-293 cells, SK cells, and mouse primary neuron (mPN) cells were pretreated with different concentrations of sunitinib or DMSO for 1 h at 37 °C before infection with ERA-mCherry at a multiplicity of infection (MOI) of 0.01. As shown in Figure 2, sunitinib significantly inhibited ERA-mCherry infection and spread in HEK293 cells, SK cells, and mPN cells in a dose-dependent manner (Figure 2A–F). The 50% inhibitory dose was approximately 2 µM, 4 µM, and 3 µM in HEK293, SK, and mPN cells. Sunitinib had a very minor effect on the cell viability of HEK293 cells (Figure 2B) and SK cells (Figure 2D), whereas at 4 µM, it reduced the viability of mPN cells by 15% (Figure 2F). ERA-mCherry growth titers in HEK293 cells treated with 3 µM sunitinib showed a decreased viral titer at different time points post-infection (Figure 2G). We also performed a gain-of-function assay by transfecting AAK1 cDNA into HEK-293 cells. At 48 h post-transfection, the cells were treated with 2 µM sunitinib for 1 h before they were infected with ERA-mCherry at an MOI of 0.01. The results showed that ERA-mCherry infection was completely restored by AAK1 cDNA transfection (Figure 2H). These results indicate that AAK1 kinase activity facilitates RABV infection.

### 3.3. Phosphorylation of AP2M1 by AAK1 Is Required for RABV Infection

AP2 functions as a cargo adaptor and links the transmembrane receptor and clathrin to form CCPs. The function of AP2 is tightly regulated by AAK1 [13,16]. Specific siRNA silencing tests showed that ERA-mCherry infection was inhibited by AP2M1 knock-down in HEK293 cells (Figure 3C,D) and SK cells (Figure 3E,F). These results demonstrated that AP2M1 is required for RABV infection. Phosphorylation of AP2M1 threonine 156 (T156) by AAK1 is essential for the activation of AP2 and may enhance cargo binding [17,18]. Therefore, we tested if the phosphorylation of AP2M1 was enhanced upon RABV infection. GAK belongs to the Prk/Ark kinase family, phosphorylates the clathrin adaptor complexes AP1 and AP2, and plays a role in CME [19]. Erlotinib binds GAK with high affinities [15]. As shown in Figure 4G, the phosphorylation of AP2M1 was significantly enhanced upon ERA-mCherry infection, and the phosphorylation was effectively inhibited by sunitinib but not erlotinib. Although it has been reported that GAK can phosphorylate AP2, our results here showed that RABV-induced AP2 phosphorylation was not mainly caused by GAK.

To further demonstrate that AP2M1 phosphorylation is essential for RABV infection, we transfected the cDNAs of AP2M1 and its dominant-negative counterpart AP2M1(T156A) into HEK293 cells. AP2M1(T156A) cannot be phosphorylated but can block the phosphorylation of endogenous AP2M1 [20]. Cells were inoculated with ERA-mCherry at an MOI of 0.05 at 24 h post-transfection, and viral titers in the cell supernatant were detected at different time points post-infection. The results showed that the growth of ERA-mCherry in AP2M1 (T156A)-transfected HEK293 cells was significantly reduced compared with that in AP2M1- and pCAGGS-transfected cells at different time points post-infection (Figure 3H). These data demonstrate that AAK1-mediated AP2M1 phosphorylation is required for RABV infection in vitro.

We tested whether GAK is required for RABV infection. We knocked down GAK expression by transfecting HEK293 cells with GAK-targeting siRNAs; reduction of GAK protein expression at 48 h post-transfection was achieved, as shown by Western blotting (Figure 4A). Quantitative RT-PCR demonstrated that GAK mRNA expression was reduced GAK1 siRNA-transfected HEK293 cells (Figure 4B). We found that siRNA knock-down had only a minor inhibitory effect on ERA-mCherry infection in HEK293 cells (causing a 20% decrease; Figure 4C) and SK cells (causing a 10% decrease; Figure 4D). Inhibiting GAK kinase activity by using erlotinib barely inhibited ERA-mCherry infection in HEK293 (Figure 4E), SK cells (Figure 4F), and mPN cells (Figure 4G). These results indicate that GAK plays a minor role in RABV infection in vitro.

### 3.4. AAK1 Functions at the Endocytosis of RABV Infection

Since AP2 plays a pivotal role in the clathrin-mediated endocytosis, we next determined whether AAK1 is required for RABV internalization. First, we performed a time-of-addition assay in HEK293 and SK cells, respectively. We found that although the addition of sunitinib inhibited RABV infection at all time points (from -1 to 9 h), but the strongest inhibitory effect appeared at -1 and 0 h post-infection, indicating that inhibition of AAK1 mainly affected the very early stage of RABV infection (Figure 5A,B).

We then tested whether the knock-down of AAK1 affected RABV binding to the cells and found that ERA-mCherry binding in the sunitinib-treated cells was comparable to that in DMSO-treated cells (Figure 5C). Next, we observed the co-localization of ERA-N-mCherry (a recombinant ERA expressing ERA *N* fused to *mCherry*) with AP2 or Rab5 in cells treated with or without sunitinib. We found that the co-localization of RABV and Rab5 was largely reduced by sunitinib treatment (Figure 5D I). The ratio of RABV: Rab5 co-localization and RABV: AP2 co-localization was significantly lower in sunitinib-treated cells than in untreated cells (Figure 5D II), indicating that the sunitinib treatment reduced the number of RABV-containing CCPs, leading to reduced RABV-Rab5 co-localization in the early endosomes. These data demonstrate that AAK1 functions at a post-receptor binding in the early stage of RABV infection and mainly affects the endocytosis of RABV.

### 3.5. Sunitinib Prolongs the Survival but Not Prevents the Death of Mice Challenged with RABV Street Virus

Since sunitinib had a significant inhibitory effect on RABV infection in vitro, we asked whether sunitinib also had anti-RABV activity in vivo. Groups of 10 mice each were given sunitinib only (to test for toxicity) and were challenged with only RABV street virus GX09 [21] or GX09 given in combination with sunitinib for five days post-challenge. Sunitinib treatment prolonged the survival of GX09-challenged mice in a statistically significant way (Figure 6). Sunitinib did not show any toxicity in the mice at the given dosage based on the observation period of 21 days. Our results suggest that AAK1 may have potential as an anti-RABV drug target.

## 4. Discussion

CME is a fundamental, well-studied process that cells use to take up extracellular nutrients, hormones, and transmembrane receptors, and also to exchange neural transmitters [22]. Clathrin adaptors are the core components of CME and they play a crucial role in regulating CCV formation. Many proteins can serve as clathrin adaptors, such as amphiphysins1 and 2, AP2 complex, β-arrestin 1 and 2, ARH protein, Dab2, and epsin 1 [8]. Other clathrin adaptors, such as AP1 complex and GGA proteins, are involved in trafficking between the TGN and endosomes. The different adaptors are preferentially involved in the CME of different cargo proteins. Many viruses are reported to use CME to invade cells, including vesicular stomatitis virus [23], influenza virus [24], adenovirus [25,26], dengue virus, and foot-and-mouth disease virus [27], with different clathrin adaptors being involved during the viral endocytosis. RABV uses multiple receptors to invade cells; to date, four proteins-nAchR, namely, acetylcholine receptor subunit alpha, CHRNA1 [28], NCAM (neural cell adhesion molecule) [29], p75NTR (low-affinity nerve-growth factor receptor) [30], and mGluR2 (metabotropic glutamate receptor 2) [12], have been proposed to play roles as receptors for RABV entry. NCAM or p75NTR are reported to be internalized via CME [31,32] in an AP2-dependent or independent way, respectively. Whether nAchR undergoes endocytosis in physiological conditions is rarely reported. The endocytosis route of the mGluR family has not yet been clarified, but one might speculate that β-arrestins could serve as adaptors during the CME of the GPCRs to which mGluRs belongs. Piccinotti and colleagues reported that a VSV-vectored chimeric virus expressing RABV G in place of VSV G (rVSV-RABVG) infects epithelial cells and mainly neurons primarily via actin-dependent CME upon receptor binding [5,33]. They found that about 24% of rVSV-RABVG internalization is AP2 independent which indicates that other adaptors may be used in the entry of RABV. Previous studies indicated that VSV, another member of Rhabdovirus, enters cells via CME, but the role of AP2 in the process remains controversial [34,35]. Interestingly, we found that RABV infection is moderately affected by AP2 knockdown, but RABV entry did induce the phosphorylation of AP2M1 by AAK1, which is required for the internalization of RABV, as sunitinib significantly inhibit RABV entry. Our study thus indicated that the endocytosis of at least one major RABV receptor (possibly be undiscovered) is AP2 dependent. Considering the possibility of the existence of as yet unidentified RABV receptors, it is conceivable that RABV enters cells via other non-clathrin dependent routes. This could also explain, in part, why the sunitinib treatment prolonged the survival time of mice but did not protect them from lethal challenge.

Interestingly, we found that AAK1 overexpression also reduced RABV replication in HEK293 cells (Figure 1H). A previous study by Conner and Schmid [36] showed that overexpression of AAK1 led to the displacement of AP2 from the membrane punctate structure, while leaving clathrin recruitment unaffected. A reduction in membrane associated AP2 could explain the observation that overexpression of AAK1 resulted in decreased RABV titers in vitro. Alternatively, this result may demonstrate the requirement of AP2 phosphorylation by AAK1 for RABV infection.

Since GAK also phosphorylates AP2M1 [19], we tested whether GAK played a role in RABV infection in vitro. GAK siRNA knock-down had a minor inhibitory effect on RABV infection in cells, whereas erlotinib barely inhibited RABV infection in cells. These results indicate that RABV entry required AP2M1 phosphorylation that is mainly caused by AAK1 but not GAK. The RABV’s preference of AAK1 over GAK during its endocytosis needs to be further studied. Both AAK1 and GAK play important roles in HCV infection; blocking GAK kinase activity by use of erlotinib significantly inhibits HCV infection [37,38]. The different requirement of GAK in HCV and RABV infection may reflect the use of different mechanisms by HCV and RABV in CME-dependent infection. Our results demonstrate that AAK1 plays a major role in RABV infection, thus further clarifying the mechanism of RABV internalization.

Sunitinib is a multi-target receptor tyrosine kinase (RTK) inhibitor that is approved for use as an anticancer drug. Sunitinib inhibits multiple RTKs, including PDGFR, VEGFR, CSF-R, and RET [15]. Sunitinib has been shown to have broad antiviral activity and potential as an antiviral against viruses, such as HCV [37,38,39], EBOV, WNV, DENV, and ZIKV [40]. In the present study, we demonstrated that sunitinib effectively inhibited RABV infection, even though sunitinib is a non-selective RTK inhibitor. Our results showed that AAK1 transfection reversed the sunitinib-induced inhibition of RABV infection (Figure 2H), indicating that sunitinib likely inhibited RABV infection by blocking the kinase activity of AAK1. Since RABV is a fatal, highly neurotropic virus, once an infection is established, death is virtually inevitable. Our mouse study showed that sunitinib could prolong the survival time of mice challenged with RABV street virus. Therefore, combining sunitinib with other drugs targeting other stages of the RABV life cycle could lead to improved rabies post-exposure prophylaxis.

In conclusion, we found that AAK1 plays an important role in RABV infection by phosphorylating AP2M1, which is essential for the clathrin-mediated endocytosis of RABV during host cell entry. Sunitinib treatment effectively inhibited RABV infection in vitro and prolonged the survival of mice challenged with RABV street virus in vivo. Understanding the role of AAK1 in RABV infection may lead to the discovery of new targets for the treatment of RABV infection.

## Figures and Tables

**Figure 1 viruses-12-00045-f001:**
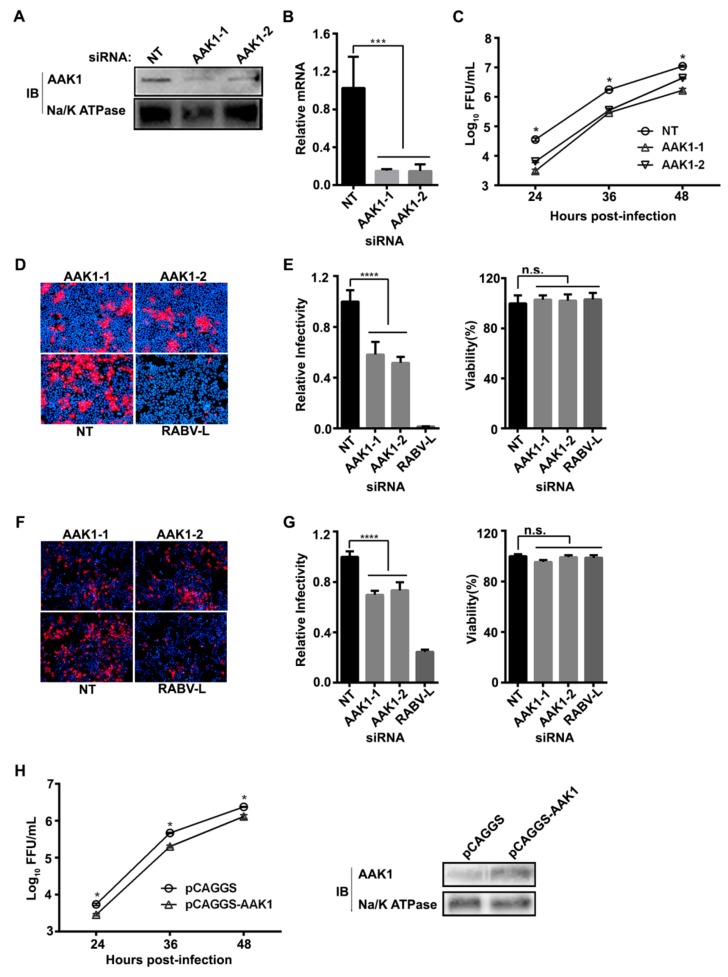
AAK1 is required for RABV infection. (**A**) HEK293 cells were transfected with siRNAs AAK1-1, AAK1-2, RABV-L, or NT. AAK1 expression level was determined by Western blotting. Numbers represent the relative ratios of AAK1 to Na/K ATPase protein normalized to the NT control. (**B**) mRNA levels of AAK1 were determined by qRT-PCR in cells followed by siRNA transfection. A one-way ANOVA was used for the statistical analysis, ***, *p* < 0.001. (**C**) Knock-down of AAK1 inhibits ERA-mCherry replication in HEK293 cells. Viral titers were determined as focus forming units (FFU) in BSR-T7/5 cells. Multiple *t tests* were used for the statistical analysis, *, *p* < 0.05. (**D**–**G**) Representative images of ERA-mCherry-infected HEK-293 cells (**D**) or SK cells (**F**) transfected with siRNAs AAK1-1, AAK1-2, RABV-L or NT. The relative infection ratio (normalized to NT siRNA-transfected cells) of RABV in HEK-293 cells (**E**) or SK cells (**G**) transfected with the indicated siRNAs (left). Cell viability was measured at 48 h after siRNA transfection (right). Data are relative fluorescence values normalized to the level of the NT control. A one-way ANOVA was used for the statistical analysis. ****, *p* < 0.0001; n.s., not significant. (**H**) The growth kinetics of ERA-mCherry (MOI = 0.05) in pCAGGS-AAK1-transfected HEK293 cells. AAK1 overexpression was confirmed by Western blotting. A one-way ANOVA was used for the statistical analysis. *, *p* < 0.05. Values represent the mean ± SD.

**Figure 2 viruses-12-00045-f002:**
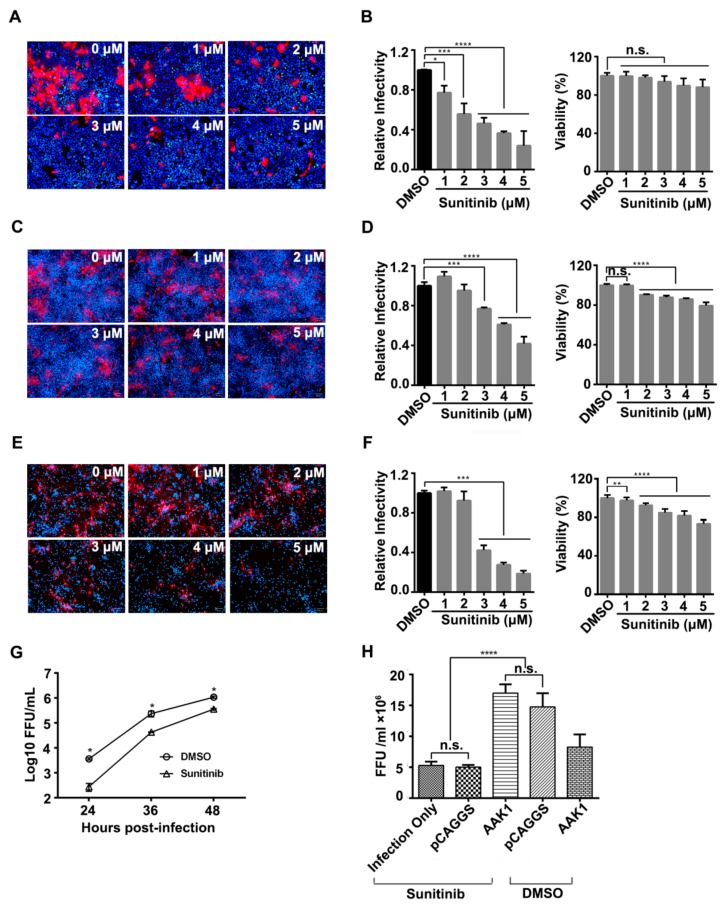
Inhibition of AAK1 kinase activity decreases RABV infection in vitro. (**A**–**F**) Cells were pretreated with different concentrations of sunitinib or DMSO for 1 h at 37 °C before infection with ERA-mCherry at an MOI of 0.01. At 48 h post-infection, a high-content quantitative image-based analysis was used to measure the relative infection ratio (normalized to NT siRNA-treated cells) of ERA-mCherry in HEK-293 cells (**A**), SK cells (**C**), and mPN cells (**E**). The viability of HEK-293 cells (**B**), SK cells (**D**), and mPN cells (**F**) treated with the indicated concentrations of sunitinib was determined using CellTiter-Glo reagent. Data are relative fluorescence values normalized to the level of the NT control. A one-way ANOVA was used for the statistical analysis. **, *p* < 0.01, ***, *p* < 0.001, ****, *p* < 0.0001; n.s., not significant. (**G**) HEK293 cells were treated with sunitinib (2μM) or DMSO and then infected with ERA-mCherry (MOI = 0.01). Viral titers were determined at the indicated time points. Multiple t-tests were used for the statistical analysis, *, *p* < 0.05. (**H**) Sunitinib counteracted the effect of AAK1 overexpression on RABV infection. HEK293 cells were transfected with pCAGGS-AAK1 and pCAGGS. At 48 h post-transfection, cells were treated with sunitinib and subsequently infected with ERA-mCherry at an MOI of 0.1. Viral titers in the supernatant were determined at 48 h post-infection and were normalized to that of the “infection only” group. Tukey’s multiple comparisons test was used to analyze the statistical difference. ****, *p* < 0.0001; n.s., not significant.

**Figure 3 viruses-12-00045-f003:**
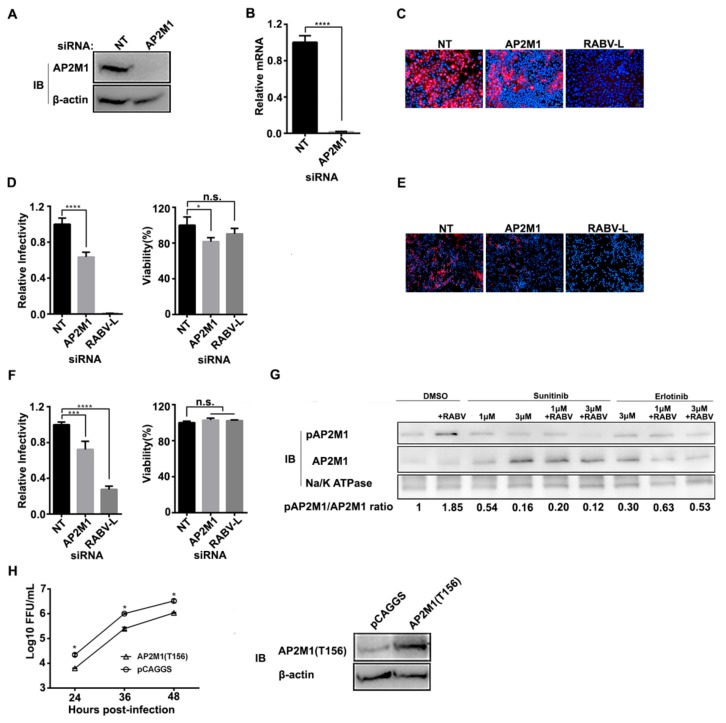
AP2M1 and its phosphorylation are important for RABV infection. (**A**) HEK293 cells were transfected with siRNAs AP2M1 or NT. AP2M1 expression level was determined by Western blotting. β-actin was used as a loading control. (**B**) mRNA levels of AP2M1 were determined by qRT-PCR in cells followed by siRNA transfection. A t-test was used for the statistical analysis. ****, *p* < 0.0001. Representative images of ERA-mCherry-infected (MOI = 0.01) HEK-293 cells (**C**) and SK cells (**E**) transfected with siRNAs targeting either AP2M1 or RABV-L, or the NT sequence. A high-content quantitative image-based analysis was used to measure the relative infection ratio (normalized to NT siRNA-treated cells) of RABV in either HEK-293 cells (**D**) or SK cells (**F**) transfected with the indicated siRNAs. Cell viability was measured at 48 h post-transfection by using the CellTiter-Glo reagent. Data are relative fluorescence values normalized to the level of the NT control. Values represent the mean ± SD. A one-way ANOVA was used for the statistical analysis. **, *p* < 0.01, ***, *p* < 0.001, ****, *p* < 0.0001; n.s., not significant. (**G**) HEK-293 cells were treated with various concentrations of sunitinib or erlotinib for 1 h at 37 °C before infection of ERA-mCherry at an MOI of 10, and then incubated for 1 h on ice to allow attachment of viral particles to cell surface receptors. Then the residual viruses were removed, and cell plasma membrane proteins were isolated to detect the phosphorylation of AP2M1 by Western blotting. (**H**) HEK-293 cells were transfected with pCAGGS-AP2M1(T156A) or pCAGGS. At 48 h post-transfection, the cells were infected with ERA-mCherry (MOI = 0.05), and virus titers were determined as focus forming units (FFU) in BSR-T7/5 cells. Multiple t-tests were used for the statistical analysis, *, *p* < 0.05. AP2M1-T156 overexpression was confirmed by Western blotting (right panel).

**Figure 4 viruses-12-00045-f004:**
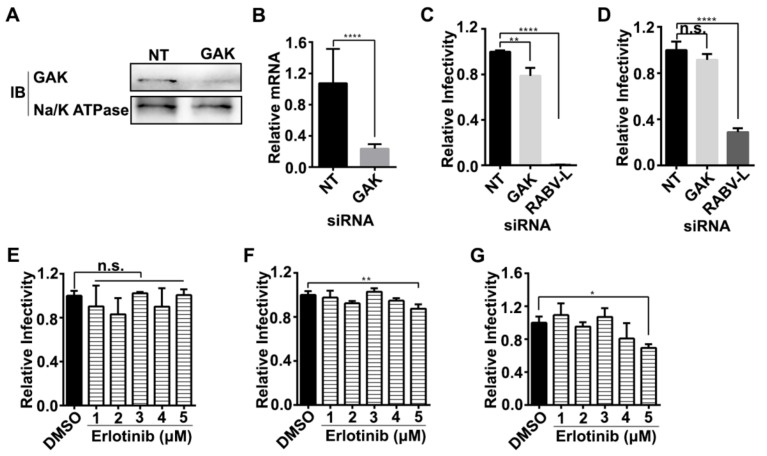
GAK has a minor effect on RABV infection in vitro. (**A**) HEK293 cells were transfected with siRNAs GAK1 or NT. GAK expression level was determined by Western blotting. (**B**) mRNA levels of GAK were determined by qRT-PCR in cells followed by siRNA transfection. A t-test was used for the statistical analysis. The relative infection ratio (normalized to NT siRNA-treated cells) of RABV in HEK-293 cells (**C**) or SK cells (**D**) transfected with siRNAs targeting either GAK or RABV-L or the NT sequence. Relative infection ratio (normalized to NT siRNA-treated cells) of HEK-293 cells (**E**), SK cells (**F**), or mPN cells (**G**) treated with erlotinib at the indicated concentrations or with DMSO for 1 h at 37 °C before infection with ERA-mCherry at an MOI of 0.01. At 48-h post-infection, a high-content quantitative image-based analysis was used to measure the relative infection ratio (normalized to NT siRNA-treated cells) of ERA-mCherry. A one-way ANOVA was used for the statistical analysis. *, *p* < 0.05, **, *p* < 0.01, ****, *p* < 0.0001; n.s., not significant.

**Figure 5 viruses-12-00045-f005:**
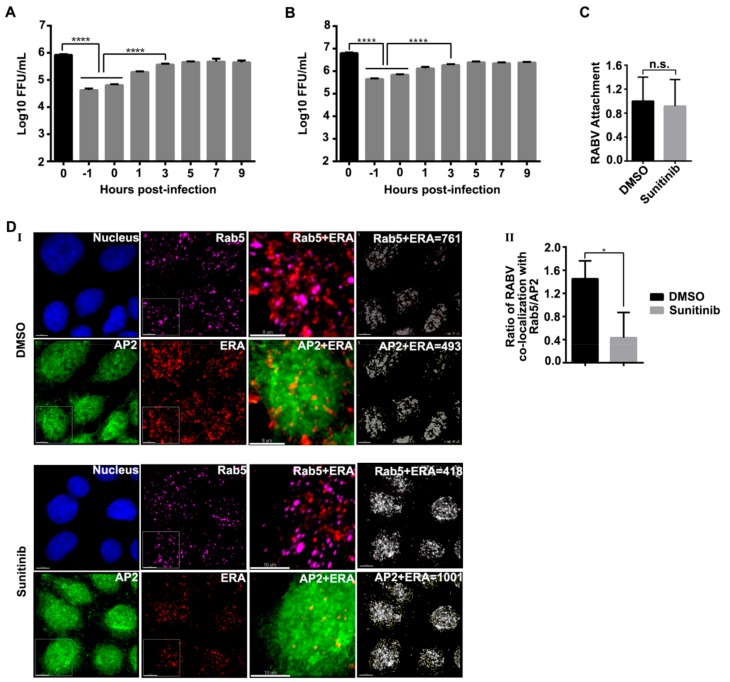
AAK1 functions at the early stage of RABV infection. HEK293 cells (**A**) and SK cells (**B**) were treated with sunitinib (3 μM) at the indicated time points before or after ERA-mCherry infection. Viral titers in the supernatant were determined at 24 h post-infection. A one-way ANOVA was used for the statistical analysis. ****, *p* < 0.0001. (**C**) HEK-293 cells were treated with sunitinib (3 μM) or DMSO for 1 h at 37 °C, and then incubated with ERA-mCherry at an MOI of 2 for 2 h at 4 °C. After removing unbound viruses by washing, total cellular RNA was extracted and subjected to qPCR to quantify cell-bound RABV RNA. A t-test was used for the statistical analysis. n.s., not significant. (**D**) Representative images showing tyramide signal amplification immunofluorescence staining of ERA-N-mCherry (red), AP2 (green), and Rab5 (purple) in sunitinib-treated or DMSO-treated cells (**I**). Co-localization of RABV with Rab5 or AP2 was observed and quantified in 15 to 20 randomly chosen cells from each sample. Statistical results represent the ratio of RABV:Rab5 co-localization normalized to RABV: AP2 co-localization (**II**). A t-test was used for the statistical analysis. *, *p* < 0.05. Bars, 8 or 10 μm.

**Figure 6 viruses-12-00045-f006:**
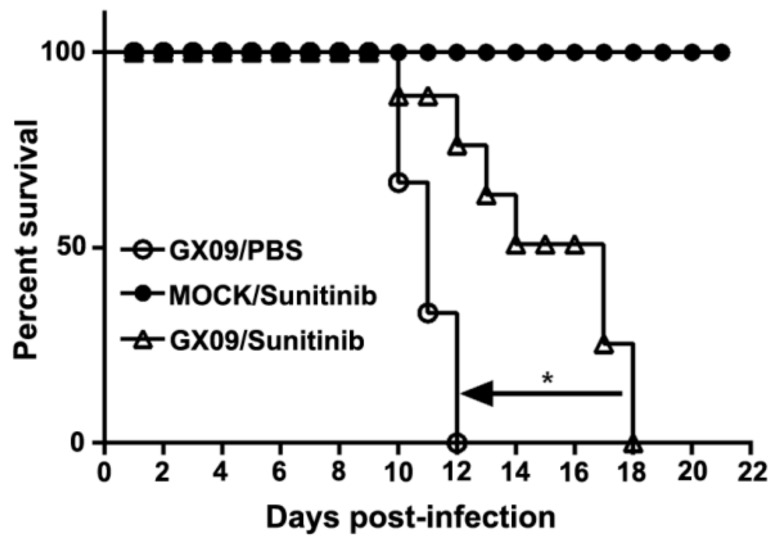
Sunitinib prolongs the survival of mice challenged with RABV street virus GX09. Three groups of ten C57BL/6 mice each, challenged with GX09 and treated with PBS (○), mock-challenged and treated with sunitinib (●), or challenged with GX09 and treated with sunitinib (△), were observed for survival post-challenge. All mice were treated once daily for five days with sunitinib or PBS. The log-rank (Mantel-Cox) test was used to analyze statistical differences between the survival rates of the challenged mice. *, *p* < 0.05; n.s., not significant.

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
