# Peer review of "The Serine/Threonine Kinase AP2-Associated Kinase 1 Plays an Important Role in Rabies Virus Entry"

_viruses, 2019, doi:10.3390/v12010045_

Round 1

Reviewer 1 Report

In their manuscript, Wang and colleagues have performed a high-throughput RNAi analysis and identified AP2-associated kinase 1 (AAK1) as an important cellular component involved in rabies virus (RABV) entry.

RABV enters the host cell through clathrin-mediated endocytosis (CME). AP2 is a complex reported as a clathrin adaptor. AP2 binds simultaneously to the cytoplasmic sorting motif of the transmembrane cargo proteinand to clathrin to form clathrin-coated pits. AP2 activity is regulated by its phosphorylation by kinase AAK1.

Using two distincts RNAs, the authors confirm that AAK1 is important for RABV infection. They show that (i) AAK1 entry facilitation is due to its ability to phosphorylate the subunit AP2M1 and that (ii) inhibition of AAK1 by sunitinib affects RABV endocytosis. Finally, they show that Sunitinib prolongs the survival but does not prevent the death of mice challenged with RABV.

Taken together, the data are convincing but I have some comments that should be taken into account.

1) Figure 3 : the quantification of the data in panels B and D do not correspond to the images presented on panels A and C, as far as the siRNA targeting L is concerned. It seems that panels A and C have been interverted.        

2) Figure 3, panel E : the migration of the band corresponding to the Na/K ATPase is very different in the mock condition compared to all the other conditions.

3) Figure 3, panel F : the immunoblot is not described at all in the legend. For AP2M1(T156A), what is the lower band ? Is it the endogeneous AP2M1 ? Does AP2M1(T156A) migrate a bit slowly than the wild type protein ?

4) in the text, the authors claim that « AAK1 is required for RABV infection » (line 152), that « AAK1 kinase activity is essential for RABV infection »  (lines 187 and 202). As the relative infectivity in presence of the siRNA directed agains AAK1 and in presence of Sunitinib is between 0.3 and 0.5, I would suggest that they tone down their conclusions by saying that AAK1 facilitates the infection.

5) The results presented here are not in agreement with those obtained on VSV (Johansdottir et al. 2009. Host cell factors and functions involved in vesicular stomatitis virus entry. J. Virol. 83:440-53) and RABV (Piccinotti et al. 2013. Uptake of rabies virus into epithelial cells by clathrin-mediated endocytosis depends upon actin. J Virol 87: 11637-11647) which does not show a strict requirement of the adaptor protein AP2. In Piccinotti et al. the authors state that « the majority (76%) of rVSV RABV G particles were seen within AP2-containing structures during entry. The remaining particles internalized in the absence of a detectable AP2 signal ». This is consistent with my previous remark (4) and, once again, this invites the authors to tone down a little bit their conclusions. In any case, these articles have to be discussed.   

 6) Line 139. The reference 3 is not the best review (neither the most recent) concerning RABV receptors. There are more recent ones: Belot et al. 2019. Structural and cellular biology of rhabdovirus entry. Adv Virus Res. 104:147-183 and Guo et al. 2019 Early events in rabies virus infection-Attachment, entry, and intracellular trafficking. Virus Research 263: 217-225. 

Reviewer 2 Report

In this manuscript, Wang et al have screened a SiRNA library and identified the Serine/threonine AP2-associated kinase 1 (AAK1) as a kinase involved in rabies virus entry in infected cells.

They have shown that :

AAK1 depletion inhibits RABV infection Inhibition of AAK1 activity by using the drug Sunitinib results in the reduction of RABV infection Phosphorylation of the adaptator protein 2 (AP2M1) threonine 156 by AKK1 is required for RABV infection The drug treatment has an effect on early step of viral infection and could affect the endocytosis of RABV In vivo experiments in mice treated with Sunitinib results in the survival of the mice.

This is an interesting and a well-written paper with appropriate experimental approaches. However some conclusions are not fully supported by the data presented, particularly the role of the AAK1 on the endocytic pathway is not convincing. Indeed the colocalisation of rab 5 / N/ AP2 is not obvious.

Different points :

The silencing of AAK1-1 is not efficient as the AAK1-1 expression after silencing is not lower that in NT RNA (Fig 1A), However, the mRNA amount appears to be reduced (Fig1B). This is surprising ! Is there any explanation ? FACS experiments could have been more appropriate to measure the relative infectivity in the different parts of the paper. How is performed the viral titration ? What is the experiment to isolate cell plama membrane proteins (Fig 3 E) ? It is not reported in the Materials and Methods. In figure 2 G the dose of the drug is not mentioned… There is a discrepency between the MOI in the Materials and Methods (MOI 0.1) and the legend (MOI 0.01) figure 2. The MOI of 0.01 is very low especially if the drug inhibits an early stage of infection.  Fig 3 A In the case of RABV L the background of the image is red. Fig 5D The colocalization is not obvious. A magnification is required to be able to observe it. A control with NH4Cl which blocks acidification and then endocytosis could have been done for comparison.

Author Response

Response to Reviewer 2 Comments

Dear Reviewer,

Thank you very much for your professional and valuable comments regarding our manuscript entitled “The serine/threonine kinase AP2-associated kinase 1 plays an important role in rabies virus entry” (656982). We believe the comments will largely help us to improve our manuscript. We have re-done some experiments and provided improved figures accordingly we made changes in the revised manuscript which are highlighted. We hope our efforts will satisfy the reviewers and meet the journal's high standards. Our point-by-point responses to the comments are as follows:

Question 1.

The silencing of AAK1-1 is not efficient as the AAK1-1 expression after silencing is not lower that in NT RNA (Fig 1A), However, the mRNA amount appears to be reduced (Fig 1B). This is surprising ! Is there any explanation ? FACS experiments could have been more appropriate to measure the relative infectivity in the different parts of the paper.

Answer.

The previous detection of AAK1 by Western blotting was done using cell membrane extract, and the qRT-PCR detection of AAK1 mRNA was done using whole cell lysate. We now have re-done western blotting using cell membrane extract, and we replaced Fig 1A. It is more convenient to detect relative infection rate using the Operetta high-content system, it has the characteristics of rapid and high-throughput, FACS experiments is costly, time consuming, and unintuitive.

Question 2.

How is performed the viral titration ?

Answer.

The method of the viral titration has been added to the Materials and Methods in line 132.

Question 3.

What is the experiment to isolate cell plasma membrane proteins (Fig 3 E) ? It is not reported in the Materials and Methods.

Answer.

The method of the cell plasma membrane protein isolation has been added to the Materials and Methods in line 136.

Question 3.

In figure 2G the dose of the drug is not mentioned. There is a discrepancy between the MOI in the Materials and Methods (MOI 0.1) and the legend (MOI 0.01) figure 2. The MOI of 0.01 is very low especially if the drug inhibits an early stage of infection.

Answer.

The dose of the drug has been added in line 236. There is a discrepancy between the MOI in the Result (MOI 0.01) and the legend (MOI 0.05) figure 2 by mistake. We have made corrections in line 237 by using the MOI=0.01. We use this assay to observe the RABV replicates in drug-treated cells, the cell number is 1.0×105 in one well of a 96-well plate, when MOI=0.01, the number of infected cells is 1.0×103, the number of infected cells would suffice for Operetta analysis.

Question 4.

Fig 3A In the case of RABV L the background of the image is red.

Answer.

Thank you for your suggestion. We have replaced the image of RABV L in Fig 3C.

Question 5.

Fig 5D The colocalization is not obvious. A magnification is required to be able to observe it. A control with NH4Cl which blocks acidification and then endocytosis could have been done for comparison. 

Answer.

We have magnified Fig 5D to make the co-localizations clearer as suggested. We think the reviewer's comments are professional and inspiring. Blocking proton entering into endosomes by ammonium chloride should affect the endosome maturation. While in the present study we showed that sunitinib treatment reduced the colocalization of RABV and Rab5 positive early endosomes which was due to fewer viruses were internalized. The cells were infected with ERA-N-mCherry (MOI = 50) at 4°C for 1 h to allow binding, then immediately shifted to 37°C for 10 min to allow internalization and observe the presence of the virus in early endosomes. During such a short time the virus has not carried out a second round of replication. From this point we think it is not necessary to add ammonium chloride at this step. Anyway, the maturation of endosomes could affact virus transport and releasing into cell plasma, we will study the question in future.

Reviewer 3 Report

Wang et al. submitted a manuscript titled “The serine/threonine kinase 1 AP2-associated kinase 1 plays an important role in rabies virus entry” for peer-review procedure in Viruses.

The authors carried out a high-through-put RNAi analysis and identified AP2-associated kinase 1 (AAK1), a serine/threonine kinase, as an important cellular component in regulating the entry of RABV. They described how AAK1 may act on RABV infection and more particularly in the early stages of infection upon entry into early endosomes. Sequentially, they studied the different partners involved in clathrin-mediated endocytosis pathway to decipher the particular role of the modulation of AAK1 in RABV infection.

A strong point of the study is that the authors have used 3 different cell types: “lab cells” (HEK-293), a more relevant cells for RABV, the neuroblastoma cells SK-N-SH and finally, the primary neurons which is the most relevant for RABV.

The virus used for all infection experiments is a recombinant virus based on ERA virus (a vaccinal rabies strain). A positive point is that the authors used a field isolate of RABV (a street isolate) to challenge the mice with the sunitinib which is more relevant than using a vaccinal strain.

Two types of recombinant virus were built: in the first one a mCherry ORF was added between the M and the G gene; in the second one a N-mCherry ORF was added between the M and the G. The author could present in supplementary the growth curves of infection to be sure that these viruses have the same growth than the parental one.

Major remarks:

-  It was not clear at all why the authors used the N-mCherry virus for figure 5D to study the co-localization of viruses with rab5. Perhaps it will be useful to add more bibliographic references allowing us to understand the reason of this construction. Furthermore, the ability of the glycoprotein to interact with rab5 was already described but the author didn’t discuss this point.

- All along the experiments, different moi were used to infect the cells. Why this heterogenicity?

- In figure 1C at 48h, the differences observed are statistically significant (p<0.05) however it is not so clear on the graph between AAK1-1 and NT.

In figure 1H, the significance is hard to believe: the viral titers obtained with or without over expression of AAK1 have less than 0.5 log of difference.

 More generally, the authors gave few or no detail on the number of replicates performed for each analysis, the number of cells analysed to measure the infectivity using the Operetta high-content system and sometimes it is difficult to be sure that some effects are biologically significant with 20-30% inhibition.

- from line 220-239, the author explained the mechanism of the phosphorylation of APM2 by AAK1 and conclude that it is required for RABV infection. They introduced some results of figure 4E using erlotinib and discuss the role of GAK in the phosphorylation of AP2.  At this stage it is difficult to understand what is GAK and what is the role of the erlotinib. We need to read Lines 257-264 to understand.  This information should be inserted before.

More generally, some controls are lacking ; it would be comfortable to have western blot analysis for all siRNA extinction (AP2M1 is not shown) and the corresponding mRNA expression as it was done in figure 1 for AAK1.

Round 2

Reviewer 3 Report

Wang et al. submitted a manuscript titled “The serine/threonine kinase 1 AP2-associated kinase 1 plays an important role in rabies virus entry” for peer-review procedure in Viruses.

The authors carried out a high-through-put RNAi analysis and identified AP2-associated kinase 1 (AAK1), a serine/threonine kinase, as an important cellular component in regulating the entry of RABV. They described how AAK1 may act on RABV infection and more particularly in the early stages of infection upon entry into early endosomes. Sequentially, they studied the different partners involved in clathrin-mediated endocytosis pathway to decipher the particular role of the modulation of AAK1 in RABV infection.

The authors clarified the different points that I had mentionned during the revewing and improved their manuscript. More informations and results are added on the different controls  which gives the reader confidence.

This article is fully ready to be published in viruses.